# ME-YOLO: Improved YOLOv5 for Detecting Medical Personal Protective Equipment

**Baizheng Wu** [1], **Chengxin Pang** [1,*], **Xinhua Zeng** [2] and **Xing Hu** [3]

1 School of Electronics and Information Engineering, Shanghai University of Electric Power, Shanghai 201306, China
2 School of Engineering and Technology, Fudan University, Shanghai 200433, China
3 School of Optical Electrical and Computer Engineering, University of Shanghai for Science and Technology, Shanghai 200093, China
* Correspondence: chengxin.pang@shiep.edu.cn

**Abstract:** Corona Virus Disease 2019 (COVID-19) poses a significant threat to human health and safety. As the core of the prevention and control of COVID-19, the health and safety of medical and nursing personnel are extremely important, and the standardized use of medical personal protective equipment can effectively prevent cross-infection. Due to the existence of severe occlusion and overlap, traditional image processing methods struggle to meet the demand for real-time detection. To address these problems, we propose the ME-YOLO model, which is an improved model based on the one-stage detector. To improve the feature extraction ability of the backbone network, we propose a feature fusion module (FFM) merged with the C3 module, named C3_FFM. To fully retain the semantic information and global features of the up-sampled feature map, we propose an up-sampling enhancement module (USEM). Furthermore, to achieve high-accuracy localization, we use EIoU as the loss function of the border regression. The experimental results demonstrate that ME-YOLO can better balance performance (97.2% mAP) and efficiency (53 FPS), meeting the requirements of real-time detection.

**Keywords:** medical personal protective equipment detection; You Only Look Once version 5 (YOLOv5); feature extraction; EIoU

## 1. Introduction

Since the outbreak of COVID-19 in 2019, the lives and health of people worldwide have been greatly threatened, causing long-term disruption to people's lives and work, and seriously hampering global economic development. The causative pathogen is a novel coronavirus (SARS-CoV-2), characterized by rapid transmission and high adaptability. Viruses are generally transmitted via droplets, contact, and aerosols, with droplet transmission being their main mode of transmission. If a patient sneezes, speaks, or spits, droplets carrying the novel coronavirus can be ejected, causing the virus to spread rapidly. Thus far, the virus has differentiated into Alpha, Beta, Gamma, Delta, and other variant strains [1]. As of 2022, 600 million people have been diagnosed cumulatively worldwide, with an average of one death from COVID-19 every 8 s [2]. Moreover, the rate of COVID-19 infection is growing exponentially worldwide, with an average daily growth rate of 1.9 times. Despite the increasing number of people being vaccinated, COVID-19 still greatly threatens people's health and safety, and the importance of protection against it cannot be underestimated [3]. As the main centers of epidemic prevention and control, hospitals, isolation hotels, and nucleic acid testing sites are crowded. As the core of COVID-19 prevention and control, the health and safety of medical and nursing personnel are extremely important, and once cross-infection occurs, it can have an incalculable impact on the prevention and control of the epidemic. Theoretically, medical personal protective equipment can isolate viruses, harmful ultra-fine dust, etc., thereby effectively preventing the spread of viruses. Therefore,

medical and nursing personnel needs to wear medical personal protective equipment such as protective suits, goggles, and masks [4]. However, due to a lack of protective awareness and insufficient supply and demand of medical personal protective equipment, many medical and nursing personnel are not required to wear medical personal protective equipment, resulting in high levels of infection. Currently, the supervision of wearing medical personal protective equipment is mainly through manual testing, which results in waste of human resources and the problem of missed and wrong detections. Therefore, with the development of computer vision (CV), automatic identification and detection of medical personal protective equipment based on deep learning are of great importance.

Traditional image detection algorithms are based on the shape and color of objects for recognition [5]. These complex algorithms have certain limitations to their application, such as a lack of sufficient robustness and error detection [6]. Deep-learning-based object detection algorithms have become popular for current research and applications, as they can overcome the limitations of traditional image detection algorithms and effectively extract object features from complex scenes. Generally, deep-learning-based object detection approaches can be divided into two categories: The first is two-stage algorithms, which are based on candidate regions, such as Fast RCNN (region-based convolutional neural network) [7] and Faster RCNN [8]. Despite the high detection accuracy of two-stage algorithms, it is difficult to increase their detection speed effectively due to the amount of computation required for extracting candidate regions. The other category is one-stage algorithms, which are end-to-end learning algorithms with high detection speed, such as SSD (single-shot multi-box detector) [9], RetinaNet [10], and YOLO (You Only Look Once) series algorithms [11–17]. These algorithms directly generate the class probability and position coordinate values of the object, obtaining the final detection result from a single inspection.

The following problems currently exist in the identification and detection of current medical personal protective equipment: there are few studies on the detection of medical personal protective equipment, and the relevant datasets are sorely lacking, while mainstream object detection models perform poorly due to the problems of overlapping and obscuring medical personal protective equipment. To address these problems, based on YOLOv5, the ME-YOLO algorithm is proposed in this article. The main contributions of this article are as follows:

1.  We propose a new medical personal protective equipment detection algorithm—ME-YOLO. Firstly, to solve the problem of poor feature extraction by the backbone network when the size of objects varies, a feature fusion module (FFM) is proposed and merged with the C3 module, named C3_FFM. Secondly, to solve the problem of the traditional up-sampling method, an enhanced up-sampling module is proposed. Thirdly, to solve the problem of slow convergence of prediction box regression in CIoU loss, EIoU loss is used as the loss function of the border regression.
2.  Compared with the other mainstream object detection algorithms, the experiments demonstrate that the ME-YOLO network structure has good detection accuracy and a high detection speed, enabling it to be applied for real-time detection.

## 2. Related Work

### 2.1. Existing Work

Most current medical personal protective equipment detection methods focus only on identifying masks. Loey et al. [18] used three datasets: Labeled Faces in the Wild (LFW), Simulated Masked Face Dataset (SMFD), and Real-World Masked Face Dataset (RMFD). They also introduced a hybrid model consisting of two parts: ResNet-50 for feature extraction and traditional machine learning algorithms for classifying whether or not masks are worn. The experimental results demonstrate that the model introduced in this paper is better than other machine learning algorithms. However, they tested their model on simulated mask datasets—not real-world mask images—and the machine learning algorithms had very poor generalization capabilities. Su et al. [19] proposed a new algorithm for mask classification and detection fusing Efficient-YOLOv3 and transfer

learning, with good results, but there was still room for improvement in terms of detection speed. Nagrath et al. [20] proposed the SSDMNV2 model for face mask detection; in this model, the authors used the SSD algorithm to detect faces in real time, using the pre-trained model MobileNetV2 to predict whether the people were wearing masks or not. Yu et al. [21] improved the YOLOv4 algorithm to achieve better results in mask recognition and standard wear detection.

To date, few studies have been conducted for the detection of gloves, goggles, protective suits, face shields, and gloves. Wang et al. [22] designed a surface defect detection system for medical gloves; it uses a dual-model detection strategy, which divides edge detection and texture detection into two steps. The experimental results demonstrated that the system has a false detection rate of less than 0.05% and a missed detection rate of less than 2%. However, the detection strategy requires high computational costs and can only detect medical gloves. Le et al. [23] designed a ski goggle defect detection and classification system. Although this system has high accuracy in defect detection and classification, it can only be applied to ski goggles. To perform multi-class personal protective equipment (PPE) compliance detection, Xiong et al. [24] proposed an extensible pose-guided anchoring framework. They then used a shallow CNN-based classifier to identify PPE. Although their proposed strategy has a higher detection accuracy, their detection categories are helmets and reflective clothing; hence, this approach cannot be used directly for the detection of medical personal protective equipment.

None of the aforementioned detection categories apply to the detection of medical personal protective equipment, and the detection models cannot balance detection speed and detection accuracy well. Therefore, in order to handle these issues, we used ME-YOLO-based medical personal protective equipment detection to detect five types of medical equipment: suits, face shields, gloves, goggles, and masks. In addition, our proposed ME-YOLO algorithm can better balance performance and efficiency, enabling it to meet the requirements of real-time detection.

### 2.2. YOLOv5 Network Structure

In June 2020, UltralyticsLLC proposed the YOLOv5 algorithm. Compared with YOLOv2, YOLOv3, and YOLOv4, YOLOv5 is smaller and more convenient, enabling flexible deployment and more accurate detection.

As shown in Figure 1, the YOLOv5 network structure consists of three main parts: backbone, neck, and output. In YOLOv5, the backbone network mainly consists of the focus module, the wrapped convolution module, the C3 module, and the SPP module. The focus module performs slicing operations on the input images; the C3 module increases the feature representation ability of the network, reducing memory consumption and parameters while maintaining the accuracy of feature extraction. The neck network is designed to make better use of the features extracted from the backbone network; it reprocesses different-sized feature maps. In the fusion process, the structure of FPN and PAN is used. The output classifies the categories and location of the target via the feature map from the fusion of the neck network. Rather than using fully connected layers, YOLOv5 uses three $1 \times 1$ convolutional layers to predict the confidence, category probability, and prediction box coordinates of objects on three different-sized feature maps.

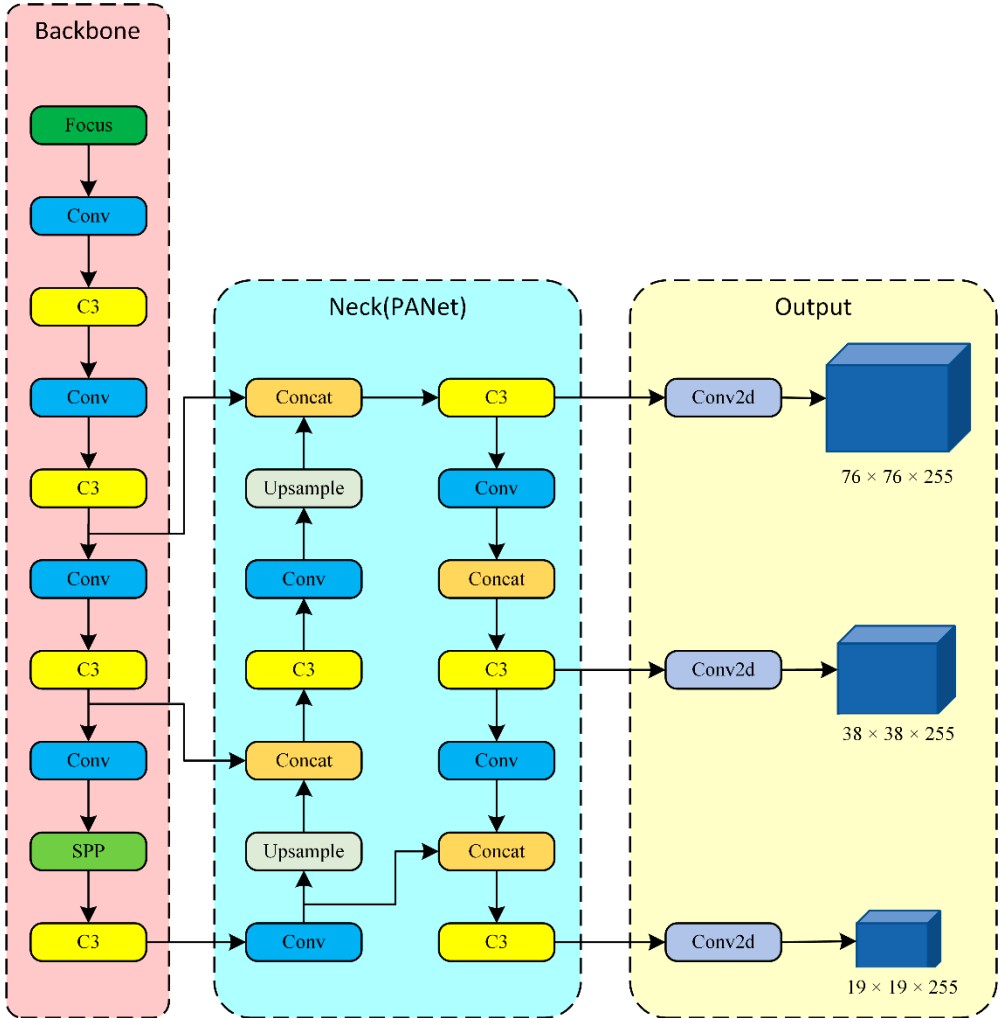

**Figure 1.** YOLOv5 network structure.

## 3. Proposed Method

As the main centers of epidemic prevention and control, hospitals, isolation hotels, and nucleic acid testing sites are crowded. Currently, deep learning-based detection of medical personal protective equipment remains challenging for three main reasons: Firstly, in complex scenes, it is difficult to distinguish small targets that are far away from the camera. Secondly, the same region may contain more than one target, and there can be a serious overlap of targets with occlusion, which greatly increases the difficulty of detection. Furthermore, current object detection models are too computationally intensive and too large to meet implementation requirements, and they cannot be deployed on edge devices. To solve the above problems, in this paper, we propose the ME-YOLO model, which can better balance performance and efficiency.

### 3.1. C3_FFM

In complex scenarios, the problem of varying sizes of medical personal protective equipment—such as face shields, protective suits, and masks—makes the direct use of the backbone network of the YOLOv5 algorithm less effective in extracting features, which can easily lead to obvious missed and incorrect detection. Therefore, to improve the recognition and detection of medical personal protective equipment of different sizes, the feature fusion module (FFM) is proposed and combined with the C3 module, named C3_FFM. The feature fusion module (FFM) expands the width of the network laterally by employing a multi-branch structure composed of different-sized convolutions on multiple branches, stitched

together to form multi-channel feature maps, thereby enhancing the receptive field of the network and its adaptability to different-sized medical personal protective equipment.

The core of the FFM is a layer-by-layer concatenated convolutional structure. As shown in Figure 2, it combines $3 \times 3$ convolutions (padding = 1), $5 \times 5$ convolutions (padding = 2), and $7 \times 7$ convolutions (padding = 3) with different receptive fields layer by layer. Convolution kernels of different sizes can detect objects of different sizes; small convolutional kernels can extract localization information of objects, while large convolutional kernels can extract deep semantic information of objects and contextual information of small objects. The use of layer-by-layer parallel convolutions can enhance continuous information and improve the ability of the network to extract objects of different sizes.

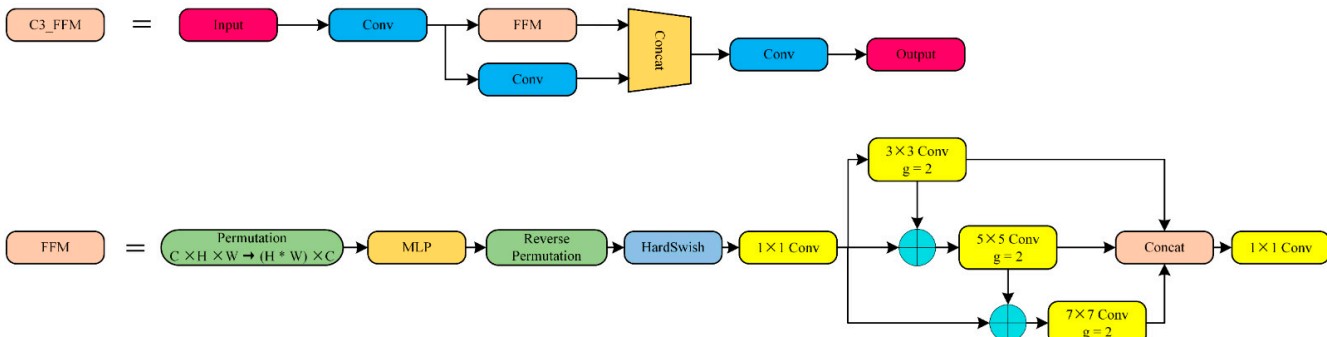

**Figure 2.** C3_FFM. The core of the Feature Fusion Module (FFM) is a layer-by-layer concatenated convolutional structure.

Firstly, we use 3D permutation to retain information across three dimensions of the input feature map, and then we use MLP to magnify the cross-dimensional channels' spatial dependencies. For efficiency, the MLP is implemented using two linear projection layers, i.e., the feature map is first downscaled by using a decay factor (r = 2) and then upscaled. The HardSwish activation function is then used to increase the non-linearity of the network, as shown in Equation (1). Secondly, to reduce the computational effort, we use $1 \times 1$ convolution to halve the size of the feature map. The feature map is then divided into 3 paths and passed through different-sized convolution kernels ($3 \times 3$, $5 \times 5$, and $7 \times 7$) in a layer-by-layer parallel manner, obtaining the feature maps of three different receptive fields. Finally, the three output channels are stitched together to retain the information of different receptive fields, and we use $1 \times 1$ convolution to change the number of channels in the output feature map, thereby obtaining the output feature map.

$$f(x) = \begin{cases} 0, & x \leq -3 \\ x, & x \geq 3 \\ x * \frac{x+3}{6}, & otherwise \end{cases} \tag{1}$$

### 3.2. Up-Sampling Enhancement Module

For medical personal protective equipment far away from the camera, the network can only extract a small number of features. Moreover, YOLOv5 uses nearest neighbor interpolation method to up-sample small feature maps, but it has small receptive fields and cannot capture rich semantic information. To address this issue, in this article, an up-sampling enhancement module (USEM) is proposed to fully retain the semantic information and global features of the up-sampled feature map. As shown in Figure 3, it consists of content-aware reassembly of features (CARAFE) [25] and an enhanced multi-head self-attention (E-MHSA) module.

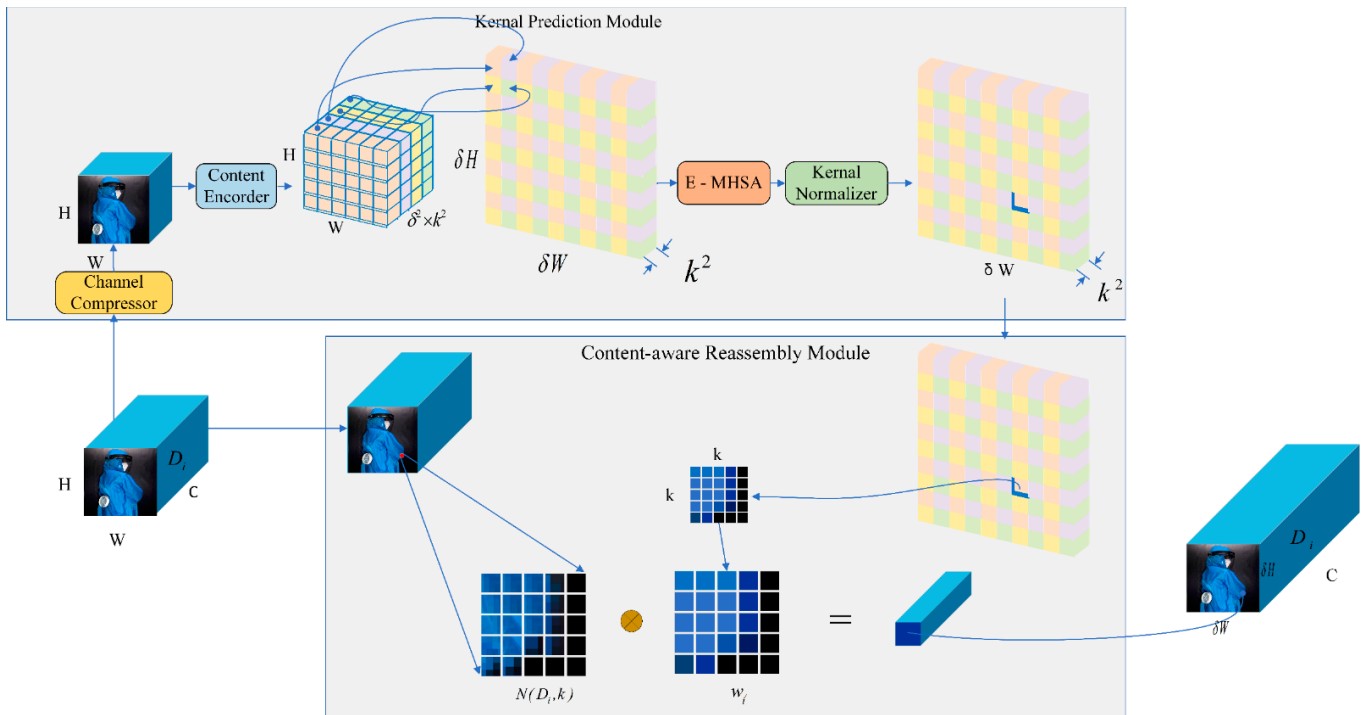

**Figure 3.** Up-sampling enhancement module (USEM). A feature map with size C × H × W is upsampled by a factor of δ (= 2) in this figure.

CARAFE can fully retain the semantic information of feature maps without adding too many parameters, thereby reducing the computational costs. CARAFE mainly consists of two modules: a content-aware reassembly module and a kernel prediction module. The role of the kernel prediction module is to generate a reconstructed convolutional kernel. Assuming that the input image size is C × H × W, where C represents the number of channels, W and H represent the height and width of the image, respectively. Firstly, to reduce the complexity, the input channels C is compressed into $C_m$ using the 1 × 1 convolution within the channel compressor module. The content encoder module then performs convolution operations on the feature map output from the previous module, generating reassembly kernels of size H × W × $K^2$ × $\delta^2$. Finally, all channels for each pixel are normalized by using softmax, so that their weights are summed to 1. The weights reflect the correlation between different channels. The content-aware reassembly module first performs the weighted sum operation on the reassembly kernel, reassembling the features within the local region. Then, each pixel of the original input feature map selects a k × k region for convolution and, finally, performs the inner product with the reassembly kernel to obtain an output feature map of size C × δH × δW.

As the CARAFE module is based on the CNN architecture, some feature map information—especially global features—will still be lost in the process of convolution [26]. Therefore, based on the Multi-Head Self-Attention (MHSA) module proposed by Srinivas et al. [27], the E-MHSA module is proposed and combined with the CARAFE module to better extract the global features of the feature map, as shown in Figure 4, where $R_w$ and $R_h$ represent the width and height of the relative position encoding, respectively, while *q*, *k*, *r*, and *v* represent the query encoding, key encoding, position encoding, and value encoding, respectively.

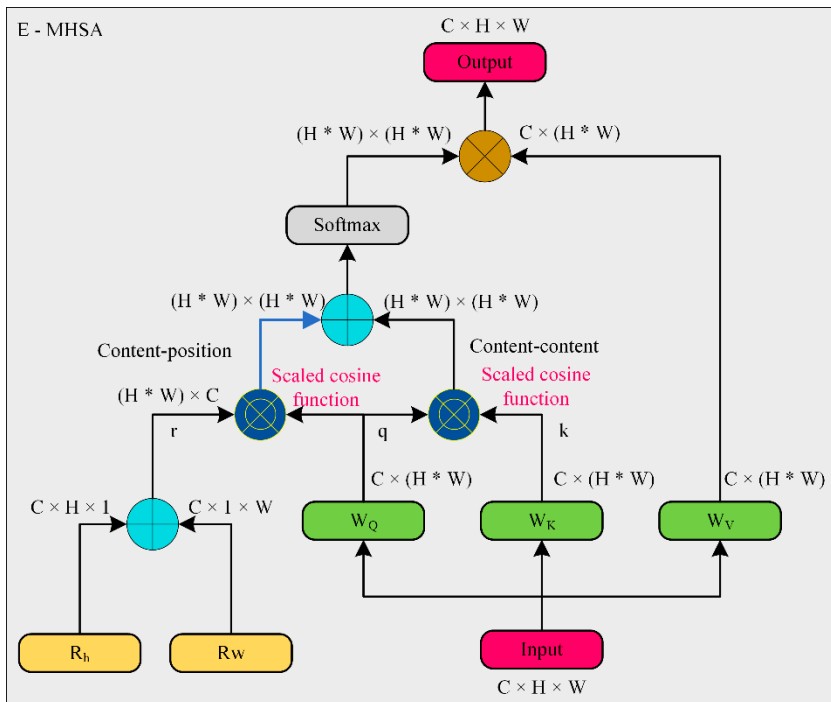

**Figure 4.** E-MHSA module. While we use 4 heads, we do not show them on the figure for simplicity.

Firstly, two learnable parameter vectors ($R_h$ and $R_w$) are initialized, and they are logically added through the broadcasting mechanism; then, the encoding of the (i, j) position is ($R_h + R_w$), and the size of the encoding matrix is (H * W) × C. Next, the input feature map of size C × H × W is passed through three independent 1 × 1 convolution and transposed to generate q, k, and v encoding matrices of size C × (H * W). In the original MHSA, the similarity between different input features is measured using dot product calculation. However, as known from [28], when using dot product calculation, certain pixel pairs can have a significant impact on the learning attention of some blocks and heads. To address this problem, in this article, we use the scaled cosine function instead of the traditional dot product calculation. As shown in Equations (2) and (3), firstly, *q*, *r* and *q*, *k* are passed through the scaled cosine function to generate the content–position and content–content encoding matrices of size (H * W) × (H * W), respectively, and then they are logically added and sent to the softmax function to generate a similarity feature matrix of size (H * W) × (H * W). Finally, using the dot product calculation with the value encoding matrix, an output feature map of size C × H × W is generated.

$$Content - position = \frac{\cos(q, \, r)}{\alpha} \tag{2}$$

$$Content - content = \frac{\cos(q, \, k)}{\alpha} \tag{3}$$

### 3.3. Improvement of the Loss Function

As the loss function can determine the difference between the prediction of the model and the actual object, it is crucial in the training process. Choosing a suitable loss function can speed up the convergence of the model and help to obtain a better model. The principle of the traditional IoU loss [29] function is 1 minus the ratio of the intersection of the prediction box and the ground-truth box to the concatenation of the prediction box and the ground-truth box. The traditional IoU loss function is formulated as follows:

$$IoU = \frac{|\, B \, \cap \, B_i \,|}{|\, B \, \cup \, B_i \,|} \tag{4}$$

$$L_{\text{IoU}} = 1 - \frac{|B \cap B_i|}{|B \cup B_i|} \tag{5}$$

where, $B$ is the area of the prediction box and $B_i$ is the area of the ground-truth box. The IoU loss function has the advantages of symmetry, non-negativity, and homogeneity, but when the ground-truth box and the prediction box are not intersected, the IoU is 0 and the model cannot continue to learn.

To avoid the above issue, YOLOv5 is based on a CIoU loss for border regression. The CIoU loss function is formulated, as follows:

$$L_{\text{CIoU}} = 1 - \text{IoU} + \frac{\rho^2\left(b, b^{gt}\right)}{c^2} + \alpha v \tag{6}$$

where $b$ is the centroid of the prediction box, $b^{gt}$ is the centroid of the ground-truth box, $\rho$ denotes the Euclidean distance between them, $c$ is the length of the diagonal of the smallest outer rectangle formed by the intersection of the prediction box and the ground-truth box. $\alpha$ is a learnable weight parameter, $v$ is the aspect ratio, and $\alpha$ and $v$ are defined, as follows.

$$\alpha = \frac{v}{(1 - \text{IoU}) + v} \tag{7}$$

$$v = \frac{4}{\pi^2}\left(\arctan\frac{w^{gt}}{h^{gt}} - \arctan\frac{w}{h}\right)^2 \tag{8}$$

Although the CIoU loss function introduces the aspect ratio of the border as a penalty term in the loss function, which continuously makes the prediction box close to the ground-truth box through iteration, it is too complicated to measure the aspect ratio, and the following two issues reduce the speed of convergence of the regression of the prediction box: (1) The difference in aspect ratio cannot fully reflect the true difference in width and height. When $h = h^{gt}$ and $w = w^{gt}$, the penalty term no longer works. (2) When the aspect ratio of the prediction box and the ground-truth box is linear, the width and height of the prediction box cannot change at the same time when regressing.

To solve the above problems, in this paper, we introduce the EIoU loss function [30], which takes into account the centroid distance and the aspect ratio. As shown in Equation (9), the EIOU loss function consists of three components: loss of overlapping area, loss of distance to the central point and loss of aspect ratio. The first two losses continue to follow the CIoU method, while the addition of the aspect ratio loss solves the problem of the CIoU loss where the length and width cannot be increased or decreased at the same time. Compared with the CIoU loss function, the EIoU loss function converges faster and with more accurate regression.

$$L_{\text{EIoU}} = L_{\text{IoU}} + L_{dis} + L_{asp} = 1 - \text{IoU} + \frac{\rho^2\left(b, b^{gt}\right)}{c^2} + \frac{\rho^2\left(w, w^{gt}\right)}{c_w^2} + \frac{\rho^2\left(h, h^{gt}\right)}{c_h^2} \tag{9}$$

### 3.4. ME-YOLO Network Structure

The framework and implementation details of the medical personal protective equipment detection model ME-YOLO are shown in Figure 5. Specifically, C3_FFM is proposed to improve the feature extraction capability of the backbone network, the up-sampling enhancement module (USEM) is proposed to fully retain the semantic information and global features of the feature map after up-sampling, and the EIoU loss is introduced as the loss function of the border regression to improve the regression convergence speed of the prediction box.

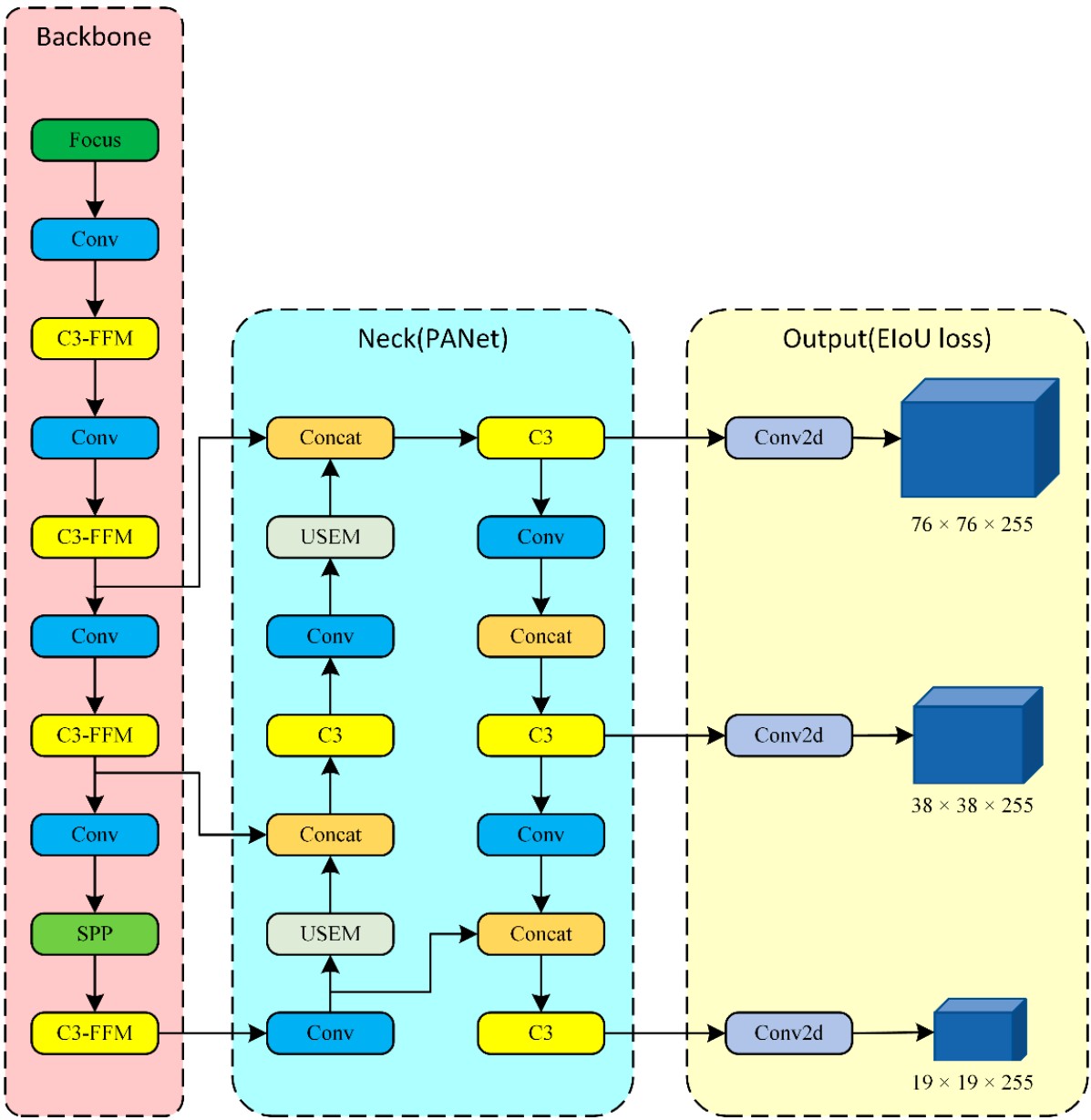

**Figure 5.** ME-YOLO network structure.

## 4. Experiments and Results

### 4.1. Dataset and Expansion Method

As there is no relevant large dataset of medical personal protective equipment available, the dataset used in this article was derived from images screened in CPPE-5 [31] as well as from our collection of medical personal protective equipment images, ultimately obtaining 2500 images with five categories—suit, mask, glove, face shield, and goggle—as shown in Figure 6, containing different scenarios as well as small targets small targets and obscured targets. The dataset was then expanded to 5000 images by panning, rotating, cropping, and color jittering. The number of the different categories in the dataset is shown in Table 1. Finally, we used the LabelImg tool to label every face in each picture in the dataset and to determine its category and coordinate information in order to obtain the real box to annotate the dataset and save the labels in xml format. As the YOLO series algorithms load tag files in txt format, the format of the tag files needed to be converted to txt format.

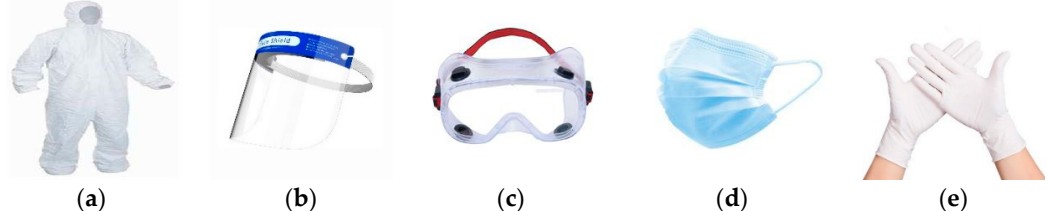

(**a**)       (**b**)       (**c**)       (**d**)       (**e**)

**Figure 6.** Categories of medical personal protective equipment. (**a**) Suit; (**b**) Face shield; (**c**) Goggle; (**d**) Mask; (**e**) Glove.

**Table 1.** The number of different categories.

| Suit | Face Shield | Goggle | Mask | Glove |
| --- | --- | --- | --- | --- |
| 10,101 | 3298 | 4133 | 10,188 | 9937 |

As shown in Figure 7, we visualized the distribution of object box occurrences and sizes in the dataset. Figure 7a represents the position of the object boxes' centroids in the image after normalizing the image sizes, and it can be observed that the objects are mostly concentrated in the center of the images; Figure 7b represents the ratio of the size of object boxes to the size of images, where it can be observed that the objects vary in size.

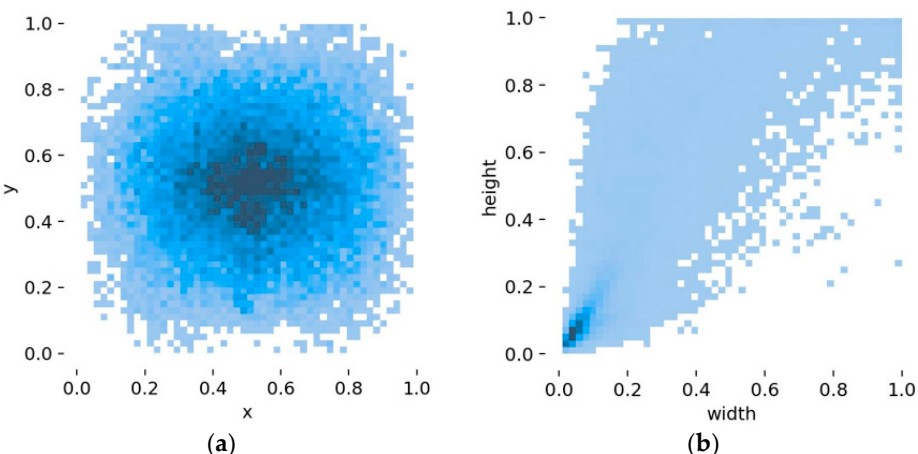

(**a**)             (**b**)

**Figure 7.** Visualization of the dataset. (**a**) The location of object boxes; (**b**) The size of object boxes.

In addition, as shown in Figure 8, the mosaic data enhancement method was applied to the images, where four training images were blended into one image by random scaling and cropping operations. Through this method, the negative impact of large objects on the detection effect of the model can be reduced, and the effect of the model in detecting small objects can be enhanced, effectively solving the problem of detecting small objects in the dataset.

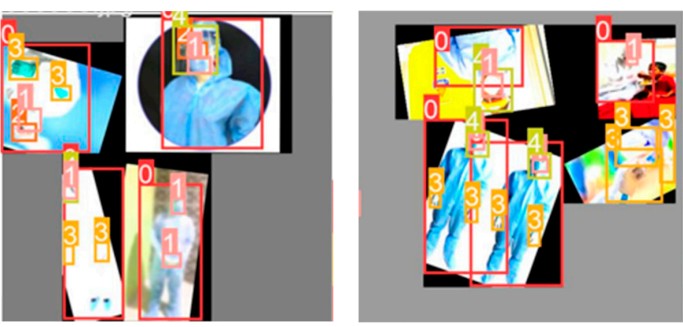

**Figure 8.** Mosaic data enhancement method.

### 4.2. Experimental Environment

The configuration parameters of the hardware and software platforms for the algorithms implemented in this article are shown in Table 2.

**Table 2.** Configuration parameters.

| Device | Configuration |
| --- | --- |
| System | Ubuntu18.04 |
| CPU | Intel®Xeon E5-2680 v4@2.40 GHz |
| GPU | GeForce RTX 2080Ti, 12G |
| GPU accelerator | CUDA 11.2, Cudnn 11.0 |
| Frames | PyTorch |
| Compilers | PyCharm, Anaconda |
| Python version | 3.6 |

For the medical personal protective equipment dataset used in this article, suitable prior boxes need to be set up to obtain accurate prediction results. As shown in Table 3, the dimensions of the prior box were calculated by using the k-means algorithm in this article. Small feature maps are suitable for detecting large objects due to the large receptive field and the content of abstract information. Meanwhile, large feature maps are suitable for detecting small objects due to the small receptive field and the content of rich positional information. Before training, the dataset was divided into a training set, validation set, and test set at a ratio of 8:1:1. In this study, we used the default hyperparameters, that were obtained by the authors of YOLOv5 for training the COCO dataset, the number of epochs was set to 2000, the batch size was set to 16, and the initial learning rate was set to 0.01.

**Table 3.** The size of prior boxes.

| Feature Map | Receptive Field | Prior Box Size |
| --- | --- | --- |
| $19 \times 19$ | Large object | $(163 \times 214)$ <br> $(179 \times 501)$ <br> $(365 \times 601)$ |
| $38 \times 38$ | Medium object | $(53 \times 81)$ <br> $(82 \times 133)$ <br> $(93 \times 61)$ |
| $76 \times 76$ | Small object | $(20 \times 31)$ <br> $(30 \times 60)$ <br> $(46 \times 43)$ |

### 4.3. Evaluation Metrics

In this study, we used six evaluation metrics to evaluate the performance of ME-YOLO: precision (P), recall (R), mean average precision (mAP), F1-score, gigaflops per second (GFLOPS), and frames per second (FPS). Precision (P) indicates the number of correctly detected samples as a percentage of the total detected samples, which can reflect the classification ability of the model for the object. Recall (R) indicates the number of correctly detected samples as a percentage of all real samples, which can reflect the detection ability of the model for the object. AP is the average of the precision at different recall, while mAP is the average of AP under all categories, which can reflect the overall performance of the model. The F1-score combines the results of the precision and recall, and model performance is proportional to the F1-score. GFLOPS means one billion floating point operations per second—the smaller the better. The formulae are as follows:

$$P = \frac{TP}{TP + FP} \tag{10}$$

$$R = \frac{TP}{TP + FN} \tag{11}$$

$$AP = \int_0^1 P(R)dR \qquad (12)$$

$$mAP = \frac{1}{Q} \sum_{q \in Q} AP(q) \qquad (13)$$

$$F1 = \frac{2 \times P \times R}{P + R} \qquad (14)$$

where TP is the number of positive samples predicted by the model as positive, FP is the number of negative samples predicted by the model as positive, and FN is the number of positive samples predicted by the model as negative.

### 4.4. Training Results and Analysis

The mAP and PR curves are shown in Figures 9 and 10, respectively. It can be observed that the mAP tends to increase as the training proceeds, while the PR curve is very smooth, with no spikes during the iterations. When reaching 300 epochs, the mAP tends to be stable. After completing 2000 epochs, the final training model is obtained.

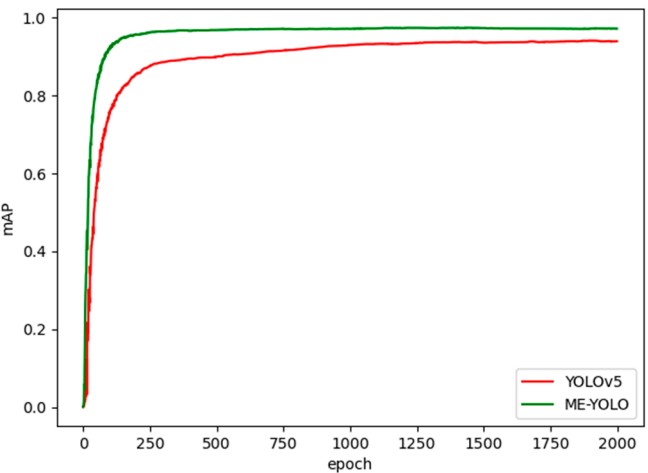

**Figure 9.** mAP.

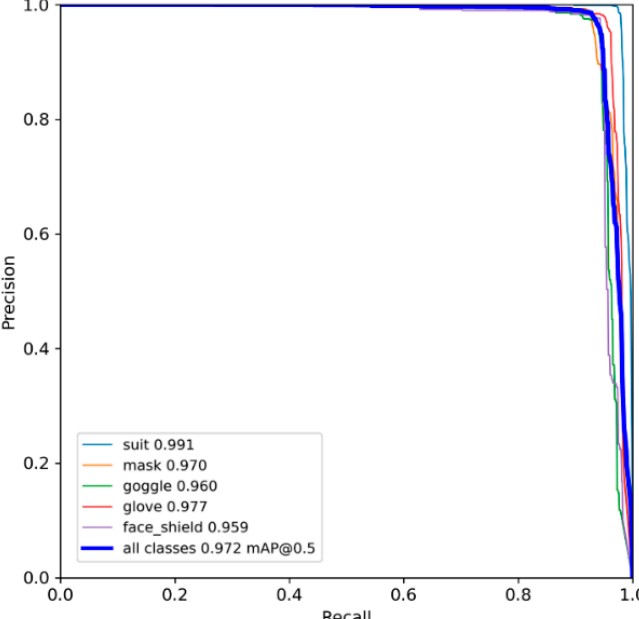

**Figure 10.** PR curve. The larger the area under the PR curve, the better the performance of the model.

### 4.5. Comparison of Mainstream Object Detection Models

To verify the performance of this algorithm for medical personal protective equipment detection, nine other network structures—SSD, RetinaNet, CenterNet, YOLOv3, YOLOv4, YOLOv4-tiny, YOLOv5s, YOLOv5m, and YOLOv5l—were selected for comparative experiments. In this experiment, the mAP, F1-score, parameters, and FPS (frames per second) were used as the evaluation metrics for the above detection algorithm. The comparative results for performance metrics achieved by the ME-YOLO algorithm and its counterparts are shown in Table 4 and Figure 11.

**Table 4.** Comparison of mainstream object detection models.

| Models | Backbone | mAP (%) | F1 | Parameters (M) | FPS (Frame·s$^{-1}$) |
|---|---|---|---|---|---|
| SSD | VGG16 | 80.1 | 75 | 90.6 | 30 |
| RetinaNet | ResNet50 | 76.0 | 58 | 138.9 | 27 |
| CenterNet | ResNet50 | 77.4 | 67 | 124.0 | 71 |
| YOLOv3 | Darknet-53 | 90.5 | 86 | 234.7 | 25 |
| YOLOv4 | CSPDarknet53 | 92.1 | 88 | 244.0 | 22 |
| YOLOv4-tiny | CSPDarknet53 | 87.8 | 83 | 22.6 | 67 |
| YOLOv5s | CSPDarknet53 | 94.2 | 93 | 7.0 | 56 |
| YOLOv5m | CSPDarknet53 | 95.4 | 94 | 21.2 | 47 |
| YOLOv5l | CSPDarknet53 | 96.1 | 94 | 46.5 | 32 |
| ME-YOLO | - | 97.2 | 96 | 7.5 | 53 |

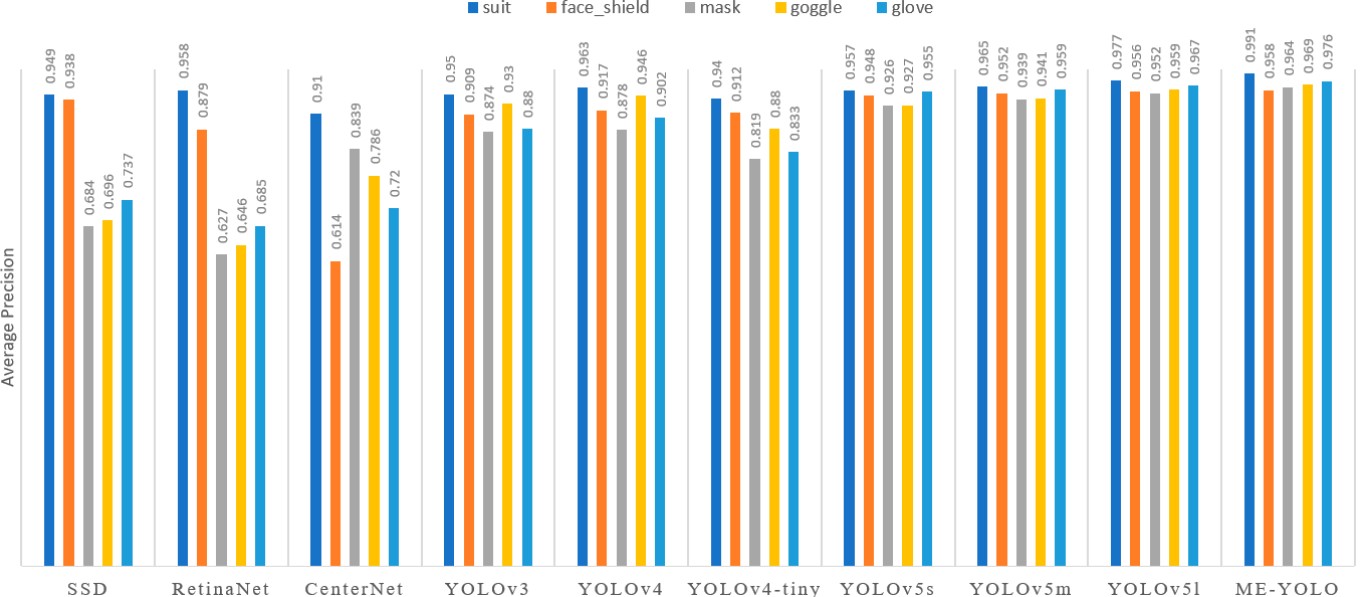

**Figure 11.** AP for each model.

To further verify the suitability of this algorithm for edge computing devices, we deployed YOLOv3, YOLOv4, YOLOv5s, YOLOv5m, YOLOv5l, and ME-YOLO on the Nvidia Jetson TX2 platform, using FPS as the evaluation metric; the results are shown in Table 5. It can be observed that the ME-YOLO algorithm achieves 42 FPS, meaning that it can be used for actual medical personal protective equipment detection tasks.

**Table 5.** Comparison of different models in FPS.

| Models | One Image Test Time(s) | All Reasoning Time(s) | FPS (Frames·s$^{-1}$) |
|--------|------------------------|-----------------------|------------------------|
| YOLOv3 | 0.047 | 23.5 | 21 |
| YOLOv4 | 0.059 | 29.5 | 17 |
| YOLOv5s | 0.022 | 11.0 | 46 |
| YOLOv5m | 0.026 | 13.0 | 38 |
| YOLOv5l | 0.037 | 18.5 | 27 |
| ME-YOLO | 0.023 | 11.5 | 42 |

Figure 12 visualizes the detection results of YOLOv5s and ME-YOLO. For dense objects, small-scale objects, and occluded objects, ME-YOLO is superior to the YOLOv5s algorithm. As can be observed from the first column of images, both ME-YOLO and YOLOv5s could detect all objects. As can be observed from the second column of images, ME-YOLO could detect all objects, but YOLOv5s mistakenly detected the scarf as a glove. As can be observed from the third column of images, ME-YOLO could detect all objects, but YOLOv5s neglected a glove. In the last column of images, YOLOv5s produced missed and false detections when the objects were severely obscured and overlapped, while ME-YOLO produced accurate detection.

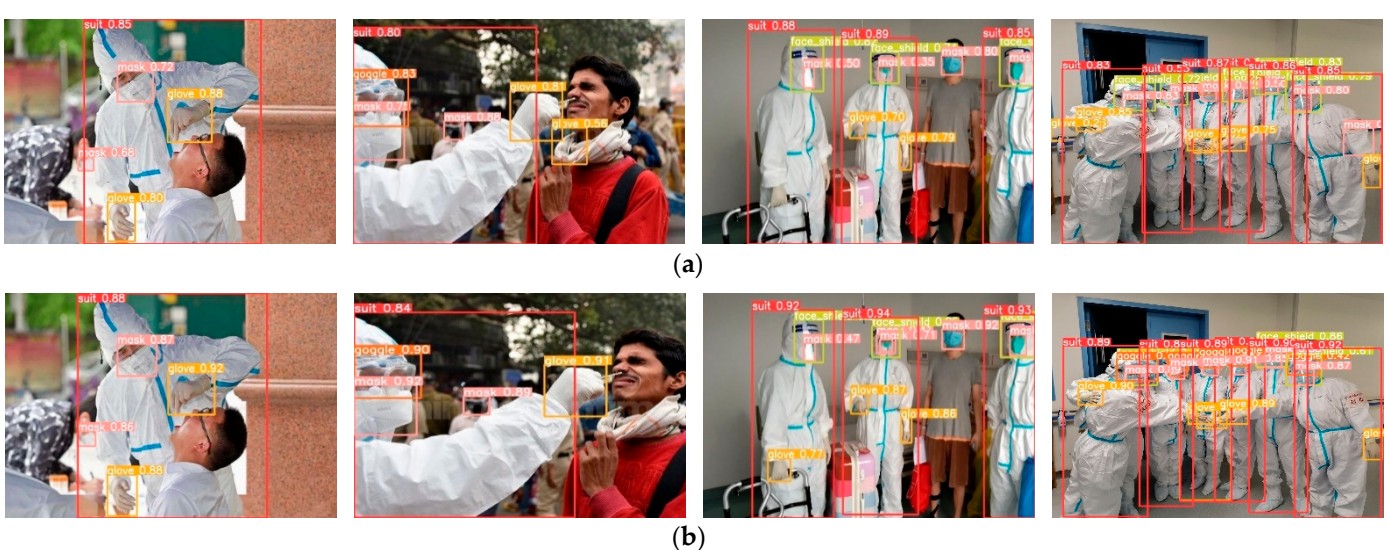

**Figure 12.** Visual comparison between YOLOv5s and ME-YOLO. (**a**) YOLOv5s; (**b**) ME-YOLO.

*4.6. Analysis of Ablation Experiments*

To further analyze the impact of different improvement methods on the performance of the YOLOv5 algorithm, three sets of experiments were designed to analyze the different improvement methods. In these experiments, precision (P), recall (R), average precision (AP), mean average precision(mAP), and F1-score were used as the evaluation metrics of experiments, as shown in Table 6, where "√" indicates that the improvement method is introduced in the model and "×" indicates that it is not introduced in the model.

**Table 6.** Ablation experiments.

| Models | C3_FFM | USEM | EIoU | AP (%) | | | | | P (%) | R (%) | mAP (%) | GFLOPS |
|--------|--------|------|------|--------|-------------|--------|------|-------|-------|-------|---------|--------|
| | | | | Suit | Face Shield | Goggle | Mask | Glove | | | | |
| YOLOv5s | × | × | × | 95.7 | 94.8 | 92.6 | 92.7 | 95.5 | 95.5 | 90.1 | 94.2 | 15.9 |
| YOLOv5s | √ | × | × | 95.6 | 94.9 | 95.3 | 95.9 | 96.6 | 96.4 | 93.4 | 95.7 | 16.8 |
| YOLOv5s | √ | √ | × | 99.0 | 95.0 | 95.5 | 96.5 | 96.5 | 97.0 | 94.3 | 96.5 | 17.5 |
| YOLOv5s | √ | √ | √ | 99.1 | 95.9 | 96.0 | 97.0 | 97.7 | 97.7 | 94.4 | 97.2 | 17.5 |

As shown in Table 6, C3_FFM can fully merge the semantic information of deep feature maps with the localization information of shallow feature maps, improving the adaptability of the backbone network to different sizes of medical personal protective equipment and its ability to extract features. Compared with the original YOLOv5s algorithm, the mAP increased by 1.5%, and the AP of the class "mask" increased significantly. In addition, GFLOPS increased by 1. Based on the above, the up-sampling enhancement module (USEM) was introduced, which can fully retain the semantic information and global features of the up-sampled feature map. Compared with the first experiment, the mAP increased by 0.8%, and the AP of the class "suit" increased significantly. The third experiment introduced EIoU loss as the loss function of the border regression, solving the problem of slow convergence of the regression of the prediction frame in the original loss function. Based on the aforementioned two experiments, the precision, recall, and mAP of the improved model were 97.7%, 94.4%, and 97.2%, respectively, i.e., 2.2%, 4.3%, and 3.0% better than the original YOLOv5s algorithm, respectively. The GFLOPS was 17.5—slightly higher than that of the original YOLOv5s algorithm.

## 5. Conclusions

In this article, we propose the ME-YOLO model, which is an improved model based on the one-stage detector. The main idea was to introduce the C3_FFM module in the backbone network to fully merge the semantic information of deep feature maps with the localization information of shallow feature maps, in order to improve the adaptability of the backbone network to different sizes of medical personal protective equipment and its feature extraction ability. In addition, an up-sampling enhancement module was introduced to fully retain the semantic information and global features of the up-sampled feature maps. Then, EIoU loss was introduced as the loss function of the border regression to improve the regression convergence speed of the prediction box. We then built a medical personal protective equipment dataset, used the k-means algorithm for clustering, and initialized the prior boxes. The performance of the ME-YOLO model was experimentally evaluated and compared to that of mainstream object detection models. The results demonstrate that the mAP of the ME-YOLO model is 97.2% and its FPS is 53 frame·s$^{-1}$, which means that this algorithm can meet the requirements for accurate and real-time detection of medical protective equipment. Due to the robustness of this deep learning algorithm for special cases such as object occlusion, it is suitable for most environments. This algorithm greatly reduces the waste of human resources and improves the efficiency of the automatic detection of medical personal protective equipment. It can be widely used in hospitals, isolation hotels, and other high-risk places, and is of great significance to protecting the health of medical and nursing personnel. In the future, the categories of medical personal protective equipment should be expanded, and the model should continue to be optimized to further improve the effectiveness of detection. Moreover, to further ensure the health of medical and nursing personnel, further research should be carried out on the identification and classification of broken protective suits.

**Author Contributions:** Conceptualization, B.W.; methodology, B.W., and C.P.; software, B.W., C.P., X.H., and X.Z.; validation, B.W.; formal analysis, B.W.; investigation, B.W.; resources, B.W., and C.P.; data curation, B.W.; writing—original draft preparation, B.W.; writing—review and editing, B.W., C.P., X.H., and X.Z.; visualization, B.W.; supervision, C.P.; project administration, C.P.; funding acquisition, C.P., X.H., and X.Z. All authors have read and agreed to the published version of the manuscript.

**Funding:** This research was funded by Science and Technology Commission of Shanghai Municipality (21JC1405300).

**Institutional Review Board Statement:** Not applicable.

**Informed Consent Statement:** Not applicable.

**Data Availability Statement:** The dataset presented in this study are available on request from the corresponding author.

**Conflicts of Interest:** The authors declare no conflict of interest.

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
