# Peer review of "ME-YOLO: Improved YOLOv5 for Detecting Medical Personal Protective Equipment"

_applsci, doi:10.3390/app122311978_

Round 1

Reviewer 1 Report

The author propose a variation of YoloV5 architecture to properly detect medical protective equipment. The approach is interesting and the result promising, however the paper contains some imprecisions and missing information.

At page 1, lines 37-40, the authors report the average COVID-19 death and contagion rate in 2022 but no reference is provided for these affirmation. The only references  related to COVID-19 are from 2020 and 2021.

At page 2, line79, the authors state that "At present, deep learning can be divided into two classes". This is not correct. There are a lot of different DL approaches. I think that the authors refer only to the most common object detection approaches.

The reference and state of the art should be revised and improved. The authors discuss only 3 articles strictly related to the topic of the paper, while the other references are very general, and some out of topic, there is no need to discuss the evolution of DL architecture from 1998. 

Page 9 line 316, which are the "other methods" used to augment the dataset? Also how many samples per class are available? Do the authors plan to make their dataset publicly available?

Table 2, It is not clear. Why is the feature map small for large object and large for small object?

Regarding the experiments it would be interesting to see which are the performance using YOLOv5m and YOLOv5l and not only YOLOv5s. 

There are also some typos in the text, e.g. "pe©rsonal" at page 2 line 91. I suggest a full grammar and spelling check.

Reviewer 2 Report

Summary:

This paper has used deep learning approach to detect medical personal protective equipments in an image. with the aim to increase the detection performance the authors have modified YoLoV5 architecture at different stages. First they added FFM to the middle of the C3 module. this module applies convolutions of different kernel sizes to the feature map from the first convolution and concatenates the outputs. Second, they replaced the nearest neighbor interpolation up-sampling approach used in YOLO-V5 with a modified CARAFE(https://doi.org/10.48550/arXiv.1905.02188) that uses MHSA before the SoftMax kernel normalizer. Last, to improve speed and localization accuracy they used a different bounding box regression objective, Efficient IOU. With these modification they have empirically demonstrated that they have outperformed other SOTA one-shot detectors and obtained superior detection accuracy with small computational overhead.

Strength

  • Given that COVID-19 has still kept to be a threat to the public safety, this kind of work can have a positive impact in preventing the spread of the virus early and strengthening the control mechanism, which will ultimately save the lives of the public and medical personnel. Specifically, the high detection accuracy achieved with out compromising the FPS makes the model convenient to be easily deployed in real time monitoring system.
  • To demonstrate the effectiveness of ME-YOLO several empirical experiments is done and compared against other related SOTA methods. In addition, ablation study is performed to analyze each modifications separately.

Weakness

  • the authors have left many details, both on the approach they proposed and the experiments, unexplained. For instance,
    1. how do you add the feature maps outputted from the different kernel size convolutions(in the FFM) ? because the different kernel size will yield feature maps with different spatial shape.
    2. it is unclear how the MLP(in the FFM) is applied to the 4D(BxHxWxC) feature map outputted from the first convolution because MLP expects 2D features
    3. on line 350-351, it says “The weights are initialized by using the pre-trained model which is trained on the COCO dataset…..”, did you train your the modified model(ME-YOLO) on COCO from scratch ? if not(if it was from YOLO-V5), how did you initialize the model parameters while you basically have a different architecture ?
  • the introduction part contains irrelevant details. in particular, line 45-75 contains too much background on image classification while the paper’s focus is on object detection. if possible, the introduction should focus on analyzing object detection approaches related to the paper instead.
  • the paper also contains wrong statements and false claims
    • line 79 states “deep learning can be divided into two classes, which are two-stage algorithms….” → this is wrong statement. it should be corrected as “Generally, deep learning based object detection approaches can be divided into two ….”
    • on line 303 the authors present EIOU as a technique introduced in this paper. However, similar objective function named EIOU is originally introduced in “https://doi.org/10.48550/arXiv.2101.08158” . this needs to be corrected immediately as falsely claiming others work is deemed as a plagiarism.
  • Apart from these technical issues, The paper needs extensive revision for language and grammar. it contains many grammatical errors and incoherent sentences which makes it hard to follow. I have tried to highlight and suggest some alternative corrections for some of the mistakes; you can find it on the attached file, the highlighted version of the paper.
  • inconsistent citation style as on line 231

Suggestion:

  • give detailed explanation on both the techniques proposed and experiments
  • remove irrelevant statements in the introduction part, replace them with background of techniques that are closely related to the paper
  • make sure to properly cite methods used from others work and use one citation style
  • edit the paper for grammar and language and explain the approaches and results clearly. if possible, Ask a third party to proof read your work.

         Additional comments can be found in attachment. Make sure to double check the highlighted content. 

Round 2

Reviewer 1 Report

The authors properly addressed my concerns and improved the text.

I think that the paper is now suitable for publication. 

Reviewer 2 Report

The author has addressed all my concerns